# a-synuclein PET Imaging: From Clinical Utility in Multiple System Atrophy to the Possible Diagnosis of Parkinson’s Disease

**DOI:** 10.3390/cells14110834

**Published:** 2025-06-03

**Authors:** Francesca Capotosti

**Affiliations:** AC Immune SA, 1015 Lausanne, Switzerland; francesca.capotosti@acimmune.com; Tel.: +41-21-345-93-41

**Keywords:** alpha-synuclein, Parkinson’s disease, multiple system atrophy, positron emission tomography

## Abstract

The development of PET tracers for the detection of pathological alpha-synuclein (a-synuclein) has the potential to revolutionize the diagnosis, monitoring, and therapeutic interventions of synucleinopathies, including Parkinson’s disease. The journey toward identifying effective PET imaging agents, however, has faced significant challenges due to the complexity and heterogeneity of the a-synuclein structures. Achieving the goal is further compounded by the low density of the pathological target, necessitating that the tracer exhibits a high binding potential, as well as the co-existence of other protein aggregates, requiring the tracer to be highly specific and selective for a-synuclein. In this perspective article, the challenges regarding developing PET tracers for a-synuclein are explored and summarized, together with the most significant recent advances in the field. These include the approaches used by our laboratories, leading to the publication of the first clinical PET images of a-synuclein pathology in patients with multiple system atrophy (MSA). Building on the current understanding of the different a-synuclein species and findings based on the success of PET tracers in the field of neurodegenerative diseases, future directions are considered also to achieve the imaging of a-synuclein pathology in Parkinson’s patients.

## 1. Introduction

In physiological conditions, a-synuclein is an intrinsically disordered protein mainly localized at the pre-synapsis, where it regulates synaptic vesicle trafficking in the neurons [1]. Under pathological conditions, a-synuclein can progressively misfold and aggregate, forming intracellular neuronal deposits known as Lewy bodies and Lewy neurites, which represent the neuropathological hallmark of Parkinson’s disease (PD) and Lewy body dementia (LBD) or glial cytosolic inclusions (GCIs) in MSA [2]. These neurodegenerative diseases associated with the accumulation of pathological a-synuclein are collectively called synucleinopathies and represent a major unmet clinical need.

To increase the probability of developing an efficacious therapeutic intervention, the accurate and early diagnosis of synucleinopathies is key, yet it remains a major challenge. While the field is moving towards a biological definition of synucleinopathies [3,4], in clinical practice, PD, LBD, and MSA are still largely diagnosed based on the symptoms presented by the patients, which appear late in the disease progression and are largely unspecific. For example, PD-associated motor symptoms usually appear 20 years after the initiation of the underlying pathology, when more than 50% of the dopaminergic neurons in the substantia nigra are already lost. Moreover, non-motor PD-associated symptoms overlap with those of other neurodegenerative conditions, such as depression, sleep disorders, neuropsychiatric disturbances, and cognitive impairment, adding complexity to achieving an accurate differential diagnosis [5].

Despite the fact that the definitive diagnosis of synucleinopathies remains only possible at post-mortem analysis, major progress has been made towards the development of biomarkers specific for a-synuclein. The most welcome advancement in the field has been related to the development of the seed amplification assay (SAA). The SAA is designed to harness the natural ability of the a-synuclein aggregates to self-replicate and then detect a very limited amount of pathological a-synuclein in vitro by employing a clever cyclic amplification reaction [6]. However, additional developmental work is still needed to exploit the use of this biomarker on a larger scale and beyond its diagnostic value. So far, the SAA acts most consistently with cerebrospinal fluid (CSF), requiring a lumbar puncture for collection, and remains a non-quantitative assay; therefore, contraindicating it as prognostic or pharmacodynamic marker. While the Food and Drug Administration (FDA) supports the use of the a-synuclein-SAA in research and clinical trials for PD, this has not yet been approved by regulatory agencies for diagnostic testing.

Employing brain PET imaging of protein aggregates, as done with amyloid plaques, is now used as a prognostic and pharmacodynamic biomarker to assess disease progression and modification. Data generated from numerous clinical trials have allowed multiple regulatory agencies to accept amyloid brain clearance captured by PET imaging as a surrogate biomarker of clinical efficacy [7].

Furthermore, tau PET imaging has proven its utility, both to support differential clinical diagnosis of tauopathies and as a screening and stratification tool in clinical trials [8]. Consequently, clinical studies are increasingly turning to these tracers to power the capture meaningful clinical benefits. It is therefore expected that the availability of PET tracers capable of imaging a-synuclein pathology in the brains of patients living with synucleinopathies will also significantly accelerate the clinical development of effective disease-modifying drugs for PD and beyond.

The search to identify a clinically useful a-synuclein PET tracer has faced multiple technical and biological challenges. The ultimate candidate must be highly brain penetrant and display high specificity and selectivity for pathological a-synuclein while being a potent binder, as sensitivity is needed for such a low-density target. This perspective article will examine the state-of-the-art of a-synuclein PET tracer development, while discussing remaining challenges and future opportunities. A particular focus will be provided concerning the first clinically validated ligand, [18F]ACI-12589 [9], and the follow-up clinical candidate displaying optimized sensitivity, [18F]ACI-15916.

## 2. Challenges in Developing Successful PET Tracers for Pathological a-synuclein

Developing PET tracers that can effectively image a-synuclein pathology in the brains of patients is a complex endeavor. One of the main reasons behind this is the fact that a-synuclein exists in multiple structural forms, each contributing differently, and often unclearly, to the disease. These forms include monomers, oligomers, fibrils, Lewy bodies, and GCIs, each possibly associated with different physio-pathological mechanisms and/or disease stages [10]. Taking this into consideration, a clinically useful a-synuclein PET tracer should specifically bind to the aggregated but not the monomeric form of a-synuclein, the latter being the physiological form of this protein.

Based on post-mortem studies, a-synuclein inclusions in the brain of patients suffering from some synucleinopathies, particularly idiopathic PD and LBD, are expected to be found in a much lower density compared to other protein aggregates, such as amyloid and tau, for which PET tracers have been successfully developed [11]. This low density of the target provokes the need to develop tracers with higher sensitivity to ultimately detect the pathology when symptoms occur or ideally, prior to this in earlier stages of disease.

Another major feature to address the development of successful a-synuclein PET tracers is the ability of a-synuclein to adopt different conformations and aggregated states, which differ among diseases. Studies using cryo-electron microscopy (Cryo-EM) have shown that a-synuclein aggregates can exist in several tertiary structures, including the Lewy fold found in PD and LBD and the MSA fold found in the so-named disease [12]. Moreover, such disease-specific conformations are extremely difficult to reproduce in vitro in test tubes [13]. The field has lost many years screening potential a-synuclein ligands on synthetic fibers only to realize that those chemical structures were not capable of binding the a-synuclein aggregates present in diseased human brain samples. Similarly, the vast majority of the a-synuclein pathology induced in animal models display very scarce resemblance to the structures of human synucleinopathies. Thus, their limited translational value restricts their utility as an intermediate step between ex vivo assessments, e.g., performed using post-mortem human tissues, and clinical evaluation. This inevitably results in a higher risk of failure in clinical development and consequently, has potentially limited the number of PET imaging candidates reaching first-in-human evaluation.

Another major challenge for the development of new PET tracers is ensuring high selectivity for the target. A successful a-synuclein PET tracer should bind selectively to a-synuclein and not cross-react with other protein aggregates, such as tau, amyloid beta, or even TDP-43. This is particularly challenging, as amyloid plaques in Alzheimer’s disease and tau tangles in tauopathies share structural similarities with the a-synuclein aggregates [14]. While difficult to achieve, selectivity is also essential because amyloidogenic protein aggerates are often present as a co-pathology in synucleinopathies. Even taking into account differences across studies, Alzheimer-related neuropathological changes were found in 20–30% of PD cases, while in cases of PD with dementia, tau and amyloid pathologies were detected in 30 and >50% of cases, respectively [15]. Interestingly, as tau and amyloid are common co-pathologies in PD, the same is true for a-synuclein as a co-pathology in AD, where it was found in >50% of cases at autopsy and shown to exacerbate cerebral glucose hypometabolism, cognitive impairment and rate of decline in AD patients [16]. This also emphasizes the utility of an a-synuclein PET tracer as a precision medicine tool, not only for the detection of a-synuclein pathology within synucleinpathies but also in other neurodegenerative diseases, including AD. The high structural similarity of the aggregates, together with the elevated frequency of co-pathologies, make selectivity a critical criterion to avoid nonspecific or inaccurate imaging results, ultimately limiting the diagnostic utility of a tracer.

Since a-synuclein pathology localizes mainly intracellularly in the brain, effective PET tracers must efficiently cross the blood–brain barrier (BBB), a selective permeability barrier that protects the brain from harmful substances. Despite the increasing evidence for BBB alterations in PD—the second most common neurodegenerative disorder with rapidly rising prevalence—with the a-synuclein accumulation possibly resulting in altered tight junction and transporter protein levels [17], the need for BBB-permeable molecules might remain an additional complication for the delivery of PET imaging agents, particularly in early disease stages [11]. This could result in a self-perpetuating pathophysiology of inflammation and BBB alteration, which contribute to neurodegeneration. This feature favors the development of small molecular weight compounds as potential imaging agents. However, these small molecules display have the desired properties for BBB and cell-membrane permeability, further restricting the explorable chemical space, e.g., in terms of physical–chemical properties.

A last, yet critical challenge to consider is the overall low resolution of PET imaging compared to the size and density of the a-synuclein aggregates. In this respect, while ensuring the high radiolabeling efficiency of the new tracers, significant efforts should also be put in place to increase the availability and accessibility of the next generation PET cameras, which offer significant improvements, including in regard to the resolution of signal [18]. Such advances are increasing the ability of a scan to produce images of much smaller amounts of a-synuclein aggregates in smaller brain areas, such as the substantia nigra, leading to the possibility to detect pathology in the earliest stages of disease.

Despite these considerations, the field has made major advancements. Until only a few years ago, even the feasibility of imaging a-synuclein pathology in the human brain was questioned; today, the goal of imaging this pathological protein has proven possible, currently in the more aggressive forms of synucleinopathy, such as MSA.

## 3. State-of-the-Art of the Development of a-synuclein PET Tracers

For many years, different laboratories focused on the identification of new hits, with only a few compounds reaching clinical testing stages. A major breakthrough came in 2023, when, for the first time, the tracer [18F]ACI-12589 finally demonstrated that (i) a-synuclein pathology can be visualized by PET and that (ii) the PET signal can differentiate MSA cases from controls and other neurodegenerative cases [9]. As summarized in Figure 1, this brain-penetrating, low molecular weight compound demonstrated high retention in the cerebellar white matter and middle cerebellar peduncles in cases of MSA dominated by cerebellar ataxia (MSA-C). The same result occurred in cases dominated by Parkinsonism (MSA-P), where tracer uptake was also observed in the lentiform nuclei. These observations correlated well with the expected distribution of the a-synuclein pathology in both MSA subtypes, a conclusion based on post-mortem data and clinical presentations [19,20]. With the publication of these first PET images of the a-synuclein pathology in patients diagnosed with MSA, the field finally had a benchmark proving the feasibility of detecting a-synuclein pathology via PET.

These landmark results have been subsequently confirmed with various compounds, which are structurally distinct from [18F]ACI-12589. [18F]SPAL-T-06 [21], [11C]MODAG-005 [22], and more recently, [18F]C05-05, a compound structurally related to [18F]SPAL-T-06 [23], have provided similar patterns of PET signal distribution as that for [18F]ACI-12589 when tested in patients diagnosed with MSA. The chemical structures of these compounds are shown in Figure 2. The fact that tracers from distinct chemical series provide similar results strongly supports the specificity of the detected signal for their common target, a-synuclein, even in the absence of a neuropathological confirmation at autopsy.

Additional data in support of the tracer specificity for pathological a-synuclein have now been provided by the first reported scan-to-autopsy case for [18F]ACI-12589 in an MSA-P case. This analysis demonstrates the overlap of the anatomical locations between the ante-mortem PET signal and the post-mortem a-synuclein pathology identified with an anti-phospho-serine 129 (pS129) monoclonal antibody. Specifically, the putamen revealed the highest PET signal retention and pS129 immunolabeled area (>3%) in the ante-mortem and post-mortem assessments, respectively. Furthermore, lower pS129 area coverage (<1%) in other brain areas, such as the middle cerebellar peduncle, also provided a minimal PET signal (unpublished data; Rodriguez et al., in preparation). These results are in line with previously reported data for [18F]Flortaucipir and tau pathology, showing that more than 1% of the area should be positive for tau pathology to result in a clear tau PET signal, particularly in smaller brain areas [24]. Taken together, these data further contribute to the growing body of evidence supporting the fact that tracers for a-synuclein also reliably and specifically image the pathology developing in MSA.

Interestingly, when [18F]ACI-12589 retention was compared between MSA cases and participants with other neurodegenerative diseases, some signals were observed in disease-affected brain areas, e.g., in AD cases. The reasons for this retention are not fully clear, and off-target binding to another neurodegenerative process or the presence of a-synuclein co-pathologies are potential, but not mutually exclusive, explanations. The premise that the observed PET signal could be, at least in part, a-synuclein-related is further supported by data showing that ACI-12589 can bind to a-synuclein co-pathology in AD tissues ex vivo, with a clean off-target profile in vitro. Some in vivo and disease-related off-target binding, however, cannot be fully ruled out at this stage [9].

Among the four a-synuclein PET tracers having shown a PET signal in MSA, [18F]ACI-12589 remains the most extensively clinically characterized. To date, >70 cases have safely received a [18F]ACI-12589 PET scan, including patients displaying idiopathic and familial forms of PD, LBD, MSA, and cases with diagnoses of other neurodegenerative disorders. Additionally, so far, this is the only a-synuclein PET tracer for which a systematic signal quantification, including kinetic modeling, was performed, providing an essential step towards clinical validation. Another key feature to differentiate [18F]ACI-12589 from the other a-synuclein PET tracers in clinical development is its selectivity. [18F]ACI-12589 is the only clinically validated PET tracer selective for a-synuclein over common co-pathologies such as amyloid beta and tau, with the other three ligands showing in vitro binding to amyloid beta and/or tau [21,22,23] and therefore, exhibiting limited utility as diagnostic agents.

Despite the extensive confirmation of the clinical utility of [18F]ACI-12589 PET for the differential diagnosis of MSA, this a-synuclein PET tracer did not show a meaningful signal in other synucleinopathies, including idiopathic PD and LBD. This differential PET signal observed within the synucleinopathy spectrum could be explained by either the higher levels of a-synuclein pathology present in MSA versus PD or LBD [25], or by the well-described differences in the conformation of the a-synuclein aggregates in these diseases [12]. Interestingly, elevated [18F]ACI-12589 retention was observed in two familial PD cases due to the a-synuclein gene (SNCA) duplication, which results in elevated a-synuclein pathology levels and an age-related development of Parkinsonian’s symptoms [26]. Particularly, both cases showed a PET signal distribution compatible with the respective clinical presentations. For instance, one of two SNCA duplication carriers exhibited mild cognitive impairment, a symptom generally associated with the spreading of the a-synuclein pathology to the cortical area, and indeed in this case, [18F]ACI-12589 PET imaging showed higher retention in the cortical areas [9]. Together, these data favor the hypothesis that the binding of [18F]ACI-12589 to pathological a-synuclein is sufficient to provide a meaningful PET signal in cases where the pathology is more abundant. However, the possibility still exists that differences in the conformation of the α-synuclein aggregates, shown by Cryo-EM as well as by SAA, also contribute to this outcome, which can be ultimately exploited in support of a differential diagnosis of MSA versus other synucleinopathies.

Based on the most likely hypothesis of the insufficient sensitivity of [18F]ACI-12589 in cases with low density a-synuclein pathology, an extensive lead optimization work was conducted to identify new structures with significantly increased binding potential. This work has resulted in the discovery of [18F]ACI-15916, a promising new a-synuclein PET imaging agent with the potential of also providing a clinically meaningful PET signal in idiopathic PD. Indeed, [18F]ACI-15916 demonstrates significantly higher specific binding to different post-mortem synucleinopathy tissues, including those in idiopathic PD cases, while retaining the good selectivity of [18F]ACI-12589 regarding probable co-pathologies (unpublished data).

In conclusion, [18F]ACI-12589 has been demonstrated a clinically useful profile towards improving the differential diagnosis of MSA, with the goal of achieving an earlier and more accurate diagnosis. Also, [18F]ACI-12589 is expected to be a valuable target engagement and/or pharmacodynamic biomarker for clinical trials in MSA.

## 4. Future Directions

Since the identification of [18F]ACI-12589, validating PET as a viable tool in the management of MSA, laboratory efforts are now focusing on the discovery of next generation compounds. The current data suggest that the features of these new compounds need to be at least as selective as, yet more sensitive than [18F]ACI-12589 to generate meaningful signals when scanning PD and LBD patients.

Beyond [18F]ACI-15916 described above, efforts in this direction include the use of [18F]C05-05. This a-synuclein tracer has shown a PET signal in five PD/LBD cases correlating with the clinical score (MDS-UPDRS-part III). The results, however, showed a substantial overlap of the signal obtained for healthy control subjects versus diseased patients, and additionally, in vitro, [18F]C05-05 binds to AD homogenates with an affinity similar or even higher than that of most of the marketed amyloid PET tracers [23]. Taking these data together, it is noted that [18F]C05-05 may not fully qualify as a reliable diagnostic marker for PD/LBD. More recently, data from two additional a-synuclein PET tracers evaluated in PD/LBD have been reported. [11C]MK-7337 showed signal retention in the substantia nigra in six of eight sporadic PD patients, but not in healthy controls, while [11C]SY08 showed tracer retention in the brainstems of PD patients, but not those of DLB patients nor controls. Both tracers, however, present substantial off-target signals in cortex, thalamus, and cerebellum or in the cerebral venous sinus network, respectively. This off-target binding, in combination with the fact that that labeling is possible only with the short half-life radioisotope, carbon-11, will significantly limit its clinical utility [27].

Currently, many laboratories are working on next generation a-synuclein tracers, and additional clinical data, including those for [18F]ACI-15916, will become available soon. It is reasonable to expect that there will be major breakthroughs and that the newly available results will be pivotal for determining whether any limitations remain for use of a-synuclein PET imaging in PD/LBD.

With these expected advances leading to the increased clinical use of a-synuclein selective PET tracers, new challenges will arise. These include the need for standardized imaging protocols to ensure consistency across clinical studies and sites. Based on findings from the field using Abeta and tau PET imaging, a-synuclein PET tracers will exhibit different profiles in terms sensitivity, specificity, off-target binding, and BBB permeability. Thus, it will be essential to establish guidelines, protocols, and ultimately, standardized quantification units, similar to the Centiloid scale, allowing for an accurate comparison between different tracers, first in clinical trials and then for routine clinical practice.

Another critical point to consider is the regulatory pathway for approval. Currently, the process is expensive and time-consuming. PET tracers targeting a-synuclein will need to undergo validation using larger sample sizes across different patient cohorts, while also being evaluated in longitudinal studies assessing their prognostic value. Discussions with regulatory agencies concerning the complexity of running scan-to-autopsy studies in slow progressing diseases, such as PD, should also be initiated.

Lastly, it would be highly desirable for the field to include a-synuclein PET imaging as a pharmacodynamic biomarker in interventional trials targeting a-synuclein. Such an endpoint, even if initially exploratory, should provide essential information concerning the efficacy of the tested therapeutic agent, as well as concerning the utility of the imaging agent for disease monitoring. Ultimately, this tool would open new avenues for faster approval of disease modification a-synuclein drugs, including the use of a-synuclein PET imaging as a surrogate biomarker of clinical efficacy, following the path delineated by the anti-amyloid drugs.

## 5. Conclusions

The development of PET tracers for a-synuclein is currently one of the most exciting areas of research in the field of neurodegenerative diseases. Available a-synuclein PET tracers, particularly [18F]ACI-12589, have shown that it is feasible to image a-synuclein pathology via PET, and these tracers are expected to transform the way in which clinicians diagnose and manage conditions such as MSA and likely, the way clinical trials in MSA are performed. Significant challenges remain in ensuring that new tracers are sufficiently sensitive, while selective, to detect pathology in idiopathic PD or LBD. As research continues to advance, the future of a-synuclein PET imaging looks more promising than ever before, with the potential to revolutionize early diagnosis, therapeutic monitoring, and personalized treatment strategies for synucleinopathies, as well as the entire neurodegenerative disease field.

## Figures and Tables

**Figure 1 cells-14-00834-f001:**
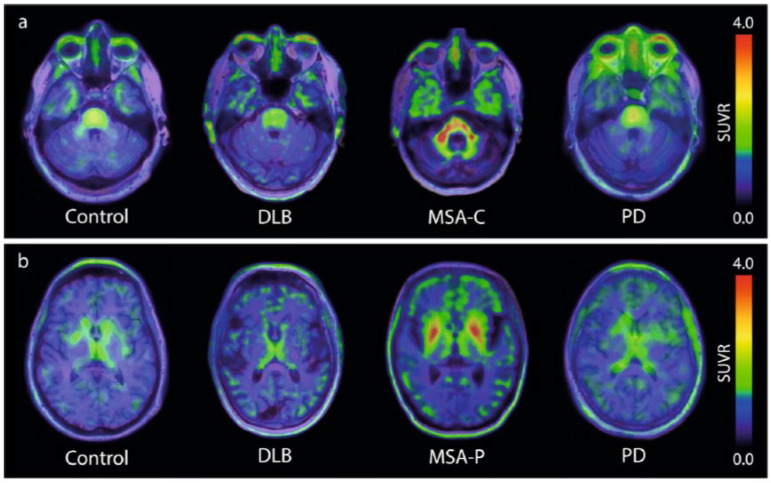
[18F]ACI-12589 PET in participants with synucleinopathies. (**a**) Presentative transversal images from [9] at the level of the middle cerebellar peduncles in a control participant, and patients with DLB, MSA-C and PD. (**b**) Representative transversal images from [9] at the level of the basal ganglia in a control participant, and patients with DLB, MSA-P and PD. SUVR images for (**a**,**b**) are averaged for the 60–90 min time frame and have been created using occipital cortex as a reference region.

**Figure 2 cells-14-00834-f002:**
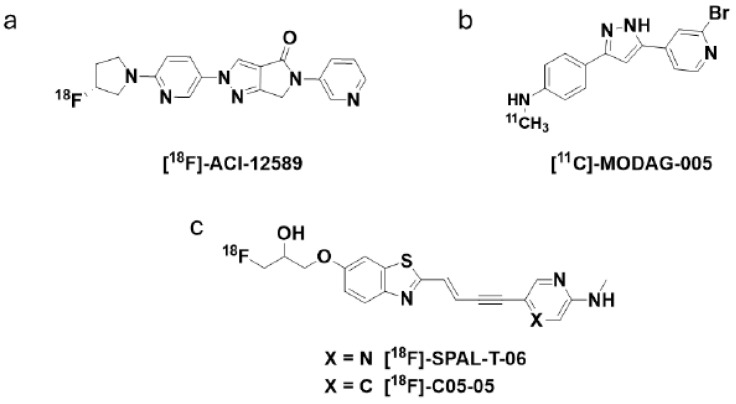
Chemical structures of a-synuclein PET ligands having shown retention in MSA cases. Chemical structures of (**a**) [18F]ACI-12589, (**b**) [11C]MODAG-005, (**c**) [18F]SPAL-T-06 and [18F]C05-05.

## Data Availability

No new data were created or analyzed in this study.

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
