# Peer review of "a-synuclein PET Imaging: From Clinical Utility in Multiple System Atrophy to the Possible Diagnosis of Parkinson’s Disease"

_cells, 2025, doi:10.3390/cells14110834_

Round 1
Reviewer 1 Report
Comments and Suggestions for Authors
This is an interesting work in developing PET tracer targeting a-synuclein. The author explained the challenges of tracer development well. The presentation is good in general. However, there are a few points that can use some improvement.
- The biggest criticism is on ACI-15916. I don't see any published data or data in the manuscript of selectivity of ACI-15917. No citation or original data can be found. I would suggest the author to limit the discussion on ACI-15916 but expand the discussion on ACI-12589.
- Page 5 line 227-229. This seems to be out of place and can be deleted.
Author Response
This is an interesting work in developing PET tracer targeting a-synuclein. The author explained the challenges of tracer development well. The presentation is good in general. However, there are a few points that can use some improvement.
I sincerely thanks the reviewer for his/her appreciation of our work and for the valuable comments provided.
- The biggest criticism is on ACI-15916. I don't see any published data or data in the manuscript of selectivity of ACI-15917. No citation or original data can be found. I would suggest the author to limit the discussion on ACI-15916 but expand the discussion on ACI-12589.
According to the provided feedback, the paragraph on ACI-15916 has been significantly shortened and simplified. Additionally, as also recommended by the editor, the reference "unpublished data" has been added. The edited paragraph (lines 216-219) now reads as follow:
Indeed, [18F]ACI-15916 demonstrates significantly higher specific binding to different post-mortem synucleinopathy tissues, including idiopathic PD cases, while retaining the good selectivity of [18F]ACI-12589 on probable co-pathologies (unpublished data).
- Page 5 line 227-229. This seems to be out of place and can be deleted.
Thanks for the comment. The entire sentence has been removed from the text., as recommended.
Reviewer 2 Report
Comments and Suggestions for Authors
Capotosti presents a communication on a-synuclein PET imaging. I have several comments regarding this manuscript:
- Author should extensively refer to off-binding properties of the radiotracer.
- The context of examination of parkinsonisms should be put in the perspective of other PET neuroimaging methods - Ref. Accumulation of Tau Protein, Metabolism and Perfusion-Application and Efficacy of Positron Emission Tomography (PET) and Single Photon Emission Computed Tomography (SPECT) Imaging in the Examination of Progressive Supranuclear Palsy (PSP) and Corticobasal Syndrome (CBS). Front Neurol. 2019;10:101. Published 2019 Feb 14. doi:10.3389/fneur.2019.00101
- Author refers to the permeability of BBB in the context of radiotracers. This issue could be evolved in the context of factors impacting the permeability in neurodegenerative disorders.
- a-synuclein PET neuroimaging may seem a striking point as differentiating feature in problematic parkinsonisms, however growing interest is associated to co-existing pathologies e.g. a-synuclein and tau. This issue could be discussed.
- A summarizing table and figure would be valuable in understanding the goal of the manuscript
Author Response
I sincerely thank the reviewer for his/her thorough review and for the valuable feedback provided, which significantly help to strengthens the message of this manuscript and to improve is quality. My point-to-point answers and clarification are provided below.
1. Author should extensively refer to off-binding properties of the radiotracer.
This is indeed an important point and an extensive paragraph to describe this has been added (lines 191-199) as follow:
Interestingly, when [18F]ACI-12589 retention was compared between MSA cases and participants with other neurodegenerative diseases, some signal was observed in disease-affected brain areas e.g. in AD cases. The reasons for this retention are not fully clear and off-target binding to another neurodegenerative process or the presence of a-syn co-pathologies [16] are potential and not mutually exclusive explanations. The premise that the observed PET signal could be, at least in part, a-syn-related is further supported by data showing that ACI-12589 can bind to a-syn co-pathology in AD tissues ex vivo and that has a clean off-target profile in vitro. Some in vivo and disease-related off-target binding, however, can’t be at this stage fully ruled out [9].
2. The context of examination of parkinsonisms should be put in the perspective of other PET neuroimaging methods - Ref. Accumulation of Tau Protein, Metabolism and Perfusion-Application and Efficacy of Positron Emission Tomography (PET) and Single Photon Emission Computed Tomography (SPECT) Imaging in the Examination of Progressive Supranuclear Palsy (PSP) and Corticobasal Syndrome (CBS). Front Neurol. 2019;10:101. Published 2019 Feb 14. doi:10.3389/fneur.2019.00101
The editor clarified that this topic will be presented in a separate article submitted to the special issue of Cells, therefore it was suggested that this comment does not need to be addressed as part of the review process for this manuscript.
3. Author refers to the permeability of BBB in the context of radiotracers. This issue could be evolved in the context of factors impacting the permeability in neurodegenerative disorders.
This is an excellent suggestion, and the following paragraph has now been added at lines 130-134:
Despite the increasing evidence for BBB alterations in PD, with the a-syn accumulation possibly resulting in altered tight junction and transporter protein levels [17], the need for BBB permeable molecules might remain an additional complication for the delivery of PET imaging agents, particularly in early disease stages [11].
4. a-synuclein PET neuroimaging may seem a striking point as differentiating feature in problematic parkinsonisms, however growing interest is associated to co-existing pathologies e.g. a-synuclein and tau. This issue could be discussed.
This is another excellent suggestion, and the following paragraph has now been added at lines 118-124:
“Interestingly, as Tau and amyloid are common co-pathologies in PD, the same is true for a-syn as co-pathology in AD, where it was found in >50% of cases at autopsy and shown to exacerbate cerebral glucose hypometabolism, cognitive impairment and rate of decline in AD patients [16]. This emphasizes the utility of an a-syn PET tracer also as precision medicine tool, not only for the detection of a-syn pathology within synucleinpathies but also in other neurodegenerative diseases, including AD.”
5. A summarizing table and figure would be valuable in understanding the goal of the manuscript
A summarizing table was considered outside the scope of this perspective article. However, to improve the clarity of some key messages and as recommended by the editor, two figures have now been added to the manuscript. Particularly, Figure 1 summarizes the key clinical data for [18F]CI-12589 in different synucleinopathy cases, including MSA-C and MSA-P cases, while Figure 2 shows the structures of the a-synuclein PET ligands mentioned in Section 3.